# Exploring the Link between Altitude of Residence and Smoking Patterns in the United States

**DOI:** 10.3390/ijerph21020226

**Published:** 2024-02-14

**Authors:** Danielle Jeanne-Marie Boxer, Young-Hoon Sung, Nicolas A. Nunez, Colleen Elizabeth Fitzgerald, Perry Franklin Renshaw, Douglas Gavin Kondo

**Affiliations:** 1Department of Psychiatry, University of Utah School of Medicine, Salt Lake City, UT 84112, USA; yh.sung@utah.edu (Y.-H.S.); nicolas.a.nunez@utah.edu (N.A.N.); colleen.fitzgerald@utah.edu (C.E.F.); perry.renshaw@hsc.utah.edu (P.F.R.); doug.kondo@hsc.utah.edu (D.G.K.); 2Department of Psychiatry and Psychology, Mayo Clinic, Rochester, MN 55905, USA; 3Intermountain Health, Oncology Clinical Trials, Intermountain Health, Salt Lake City, UT 84107, USA; 4Rocky Mountain Mental Illness Research, Education and Clinical Center (MIRECC), George E. Whalen Department of Veterans Affairs (VA) Medical Center, Salt Lake City, UT 84148, USA

**Keywords:** cigarette smoking, altitude, United States, county data

## Abstract

Introduction: Smoking-related diseases affect 16 million Americans, causing approximately 480,000 deaths annually. The prevalence of cigarette smoking varies regionally across the United States, and previous research indicates that regional rates of smoking-related diseases demonstrate a negative association with altitude. The purpose of this study was to determine the relationship between altitude and the prevalence of cigarette smoking by county (*N* = 3106) in the United States. We hypothesized that smoking prevalence among adults would be negatively associated with mean county altitude. Methods: A multivariate linear regression was performed to examine the relationship between county-level mean altitude and county smoking rate. Covariates were individually correlated with 2020 smoking data, and significant associations were included in the final model. Results: The multivariate linear regression indicated that the county-level smoking rates are significantly reduced at high altitudes (*p* < 0.001). The model accounted for 89.5% of the variance in smoking prevalence, and for each 1000-foot increase in altitude above sea level, smoking rates decreased by 0.143%. Based on multivariate linear regression, the following variables remained independently and significantly associated: race, sex, educational attainment, socioeconomic status, unemployment, physical inactivity, drinking behavior, mental distress, and tobacco taxation. Conclusions: Our results indicate that smoking rates are negatively associated with altitude, which may suggest that altitude affects the pharmacokinetics, pharmacodynamics, and mechanistic pathways involved in cigarette use. Further research is needed to explore the relationship between altitude and smoking and how altitude may serve as a protective factor in the acquisition and maintenance of tobacco use disorders.

## 1. Introduction

Cigarette smoking remains the leading cause of premature death and preventable disease in the United States. [1] Smoking is responsible for more than 6 million deaths worldwide each year, including more than 480,000 in the United States (U.S.). Moreover, exposure to secondhand smoke leads to an additional 890,000 deaths worldwide each year and causes a multitude of serious and life-threatening health problems for non-smoking adults and children [2,3]. Smoking-related diseases are primarily caused by prolonged exposure to the thousands of hazardous chemicals found in tobacco smoke, which include hydrogen cyanide, arsenic, carbon monoxide, lead, benzene, formaldehyde, and many other toxins and carcinogenic substances with harmful health consequences. [1] Research shows that direct or secondhand cigarette smoke ingestion often contributes to the development of one or more of the following: respiratory disease, including asthma, emphysema, and chronic obstructive pulmonary disease (COPD), cardiovascular disease, including hypertension, stroke, and coronary heart disease, diabetes, and neoplastic disease including cancers of the lung, liver, and colon [1].

In recent years, the negative health effects of cigarette smoke have reached public awareness through anti-tobacco advertisements and legislative initiatives aimed at curbing the use and availability of tobacco products. The Centers for Disease Control and Prevention (CDC) reports that the U.S. has experienced a decrease in tobacco use over the last decade, from 16.8% of the population in 2014 to 11.5% in 2021 [4,5]. However, in 2021, more than 35.6 million U.S. adults reported smoking tobacco products regularly, resulting in over $300 billion annually in lost productivity and direct healthcare costs [6]. In addition, the rising smoking-attributable burden of disease in low to middle socio-demographic index regions [7], and the association in youth of electronic cigarette use with greater risk for subsequent cigarette smoking initiation [8], all highlight the critical public health importance of identifying the risk factors for cigarette use.

Epidemiological data has established that the prevalence of cigarette use varies regionally across the U.S. For instance, regions with the highest smoking rates include the Midwest (22.0%) and the South (21.1%) regions, whereas states in the Northeast (16.6%) and West (15.0%) regions showed lower smoking rates [5]. This variation may potentially contribute to regional health disparities, which may include a higher prevalence of smoking-related diseases in areas with high cigarette consumption rates. Additionally, multiple factors may contribute to these smoking behaviors, such as state tobacco legislation and taxation, cultural acceptability, tobacco product availability, and access to smoking-cessation programs; in addition, environmental variables such as weather conditions and air quality have been shown to influence smoking behavior [9,10].

To date, there has been previous literature explaining an inverse association between altitude and factors such as obesity, diabetes, or metabolic syndrome [11]. Previous studies have examined the impact of altitude, smoking patterns, and physical health. A county-level ecological study by Merrill (2020) showed that obesity was primarily associated with adult smoking, lower altitude, and physical inactivity [11]. Moreover, another study aimed at examining the effects of smoking cessation on body mass utilizing Lung Health Data showed that smoking cessation was associated with an increase in body mass index mostly in younger individuals, people with no educational degrees, and those with lower BMIs at baseline [12]. Furthermore, research indicates that the altitude of residence may impact the prevalence of several mental disorders, including depression, bipolar disorder, attention-deficit hyperactivity disorder, suicide, and substance use disorders [13,14,15,16,17,18]. Specifically concerning substance use, our group previously reported that altitude of residence is positively associated with the usage of cocaine [14] and methamphetamine [19]; however, data has been inconclusive in establishing the connection between smoking and altitude in the United States.

The purpose of this study was to examine the potential relationship between altitude and the prevalence of cigarette use at the county level (*N* = 3106) across the continental United States. Published findings indicate that there is a decreased prevalence of smoking-related diseases, including lung cancer, emphysema, COPD, and asthma, in regions at higher altitudes [20,21,22]. The association between lung cancer and cigarette use is particularly strong, with 80–90% of cases linked to smoking [23,24]. These relationships suggest that altitude may also influence smoking behavior; however, there is a shortage of research examining the potential link between altitude and cigarette use in the United States. Based on previous findings indicating a negative relationship between smoking-related diseases and altitude, we hypothesized that mean county altitude would be negatively associated with prevalence of cigarette use.

## 2. Methods

### 2.1. Smoking Rate Data

County cigarette smoking rates were obtained from the 2023 County Health Ranking National Database (CHRND), which is a compilation of national statistics from 2020–2021 that includes demographics, the prevalence of physical and mental health issues, frequency of disease, as well as social and economic factors that may influence health including educational attainment level, socioeconomic status, healthcare access, and physical environment [25]. Smoking data used in the 2023 CHRND were obtained from the 2020 Behavioral Risk Factor Surveillance System (BRFSS), a health telephone survey system widely used by the CDC and other government agencies. Precisely, BRFSS data regarding cigarette use were collected from 401,958 participant interviews and smoking behavior was assessed by asking participants if they had “smoked at least 100 cigarettes in your entire life” and whether they “now smoke cigarettes every day, some days, or not at all” (Figure 1) [26]. 

### 2.2. Mean County Altitude Data

Mean county altitude data were obtained from the Shuttle Radar Topography Mission (SRTM) altitude dataset, which uses topographical radar images of the Earth to determine regional mean altitude [27]. The SRTM data were created in February 2000 and provide mean elevation calculations for each square kilometer of each county. County outlines provided by ArcGIS/ArcInfo were then overlaid on the mean spatial data to obtain the mean county elevation. Data from 3106 counties within the 48 Continental United States and the District of Columbia were used for the current analyses. SRTM digital altitude information is unavailable for Alaska and Hawaii; therefore, these states were excluded from the analyses. Data inputs for this analysis used a 1:500,000 scale U.S. counties vector dataset and a mosaiced digital elevation model of ∼0.5 km spatial resolution derived from the SRTM dataset. State elevations were calculated by taking each state’s mean of the county elevations (Figure 2).

### 2.3. Potential Covariates

Research has identified epidemiologic covariates associated with increased cigarette use, including sex, race and ethnicity, low educational attainment, rurality, unemployment, living at or below the poverty line, physical inactivity, mental distress, excessive drinking, and taxation on tobacco products [9,28,29,30,31,32,33,34,35,36,37,38]. Covariates included in this study were percentage of females, Native Americans/Native Alaskans, African Americans, non-Hispanic/whites, adults who completed high school, Americans living in rural communities, unemployment, individuals living at or below the poverty line, physical inactivity, Americans with frequent mental distress, excessive drinking, and excise tax on tobacco products. County-level data were used for each covariate except for cigarette excise tax, which is levied at the state level (*N* = 48). The covariate datasets contained 2020–2021 data and, therefore, were consistent with the timeframe of the BRFSS smoking data. The only exception was educational attainment data, which included information from 2017–2021, as this was the only county-level information available. 

County-level covariates were extracted from the 2023 CHRND, a comprehensive dataset outlining demographic and health information from 2020–2021. The CHRND is a compilation of health data collected by various U.S. government agencies, including the Census Bureau, the Centers for Disease Control and Prevention, and the Bureau of Labor Statistics [25]. County poverty rates were obtained from the 2020 U.S. Census Bureau’s Small Area Income and Poverty Estimates Program [39]. 

### 2.4. Data Analyses

Analyses of smoking rates and potential covariates were conducted to determine which variables needed to be controlled for in the model. Individual correlations were performed between smoking rates and the previously mentioned covariates. In addition to county-level altitude, predictor variables that demonstrated a significant association with county smoking rate were included in the final model. A multivariate linear regression was conducted to examine the relationship between county-level mean altitude and county smoking rate. Statistical significance for all conducted analyses was at an alpha level of 0.05. All statistical analyses were performed using Statistical Product and Service Solutions (SPSS) version 29 (IBM Corporation, Armonk, NY, USA). 

## 3. Results

This study included 3,106 counties in the United States. In Table 1, we summarize the demographics of the selected variables. Correlation analyses found that mean county altitude and all covariates showed a significant and independent association with county smoking rates (*p* < 0.05). Therefore, these thirteen variables were included in the final multivariate linear regression. The regression indicated that the overall model was significant (*F* (13, 3092) = 2039.512, *p* < 0.001). In addition, the regression model accounted for a total of 89.5% of the variance in county smoking rates (*R* = 0.946). Results from the regression model suggest that mean county altitude and all included covariates are significantly associated with U.S. county smoking rates for 2020. An inverse association was found between increased altitude and smoking (Figure 3).

Our findings suggest that for each 1000 foot increase in altitude above sea level, the county smoking rate decreased by 0.143% when controlling for other variables (β = −0.143, *p* ≤ 0.001, 95% C.I. = [−0.178, −0.108]) (Table 2). In terms of demographic variables, findings indicate that for each percent increase in Native American/Native Alaskan, African American, and non-Hispanic/white populations, smoking rates increased by 0.176% (β = 0.176, *p* ≤ 0.001, 95% C.I. = [0.167, 0.186]), 0.049% (β = 0.049, *p* ≤ 0.001, 95% C.I. = [0.043, 0.055]), and 0.113% (β = 0.113, *p* ≤ 0.001, 95% C.I. = [0.108, 0.118]), respectively. Multicollinearity diagnostics were conducted among racial categories, showing that multicollinearity was not a concern (Native American/Native Alaskan, tolerance = 0.809, VIF = 1.236; African American, tolerance = 0.547, VIF = 1.828; and non-Hispanic/white, tolerance = 0.514, VIF = 1.945). 

Moreover, the regression model suggests that smoking rates decreased by 0.070% with each percentage increase in Americans who completed high school (β = −0.070, *p* ≤ 0.001, 95% C.I. = [−0.085, −0.055]) and 0.109% with each percentage increase in female residents (β = −0.109, *p* ≤ 0.001, 95% C.I. = [−0.132, −0.087]) when controlling for all other variables. Smoking rates also increased by 0.003% for each percent increase in rurality (β = 0. 003, *p* ≤ 0.005, 95% C.I. = [0.001, 0.005]).

Results show that for each percent increase in physical inactivity, excessive drinking, and frequent mental distress, smoking rates increased by 0.410% (β = 0.410, *p* ≤ 0.001, 95% C.I. = [0.389, 0.431]), 0.078% (β = 0.078, *p* ≤ 0.001, 95% C.I. = [0.057, 0.098]), and 0.518% (β = 0.518, *p* ≤ 0.001, 95% C.I. = [0.479, 0.558]), respectively. In terms of socioeconomic variables, the regression model indicates that county smoking rates increased by 0.084% with each percentage increase in the poverty rate (β = 0.084, *p* ≤ 0.001, 95% C.I. = [0.067, 0.100]), as well as by 0.122% with each percentage increase in unemployment (β = 0.122, *p* ≤ 0.001, 95% C.I. = [0.086, 0.158]). Lastly, the results indicate that county smoking rates decreased by −0.199% with increased state taxation on tobacco products (β = −0.199, *p* ≤ 0.001, 95% C.I. = [−0.260, −0.0.138]) (Table 2).

To increase confidence in our findings, a Bonferroni correction was calculated and applied to impose a maximally conservative critical significance level of *p* ≤ 0,00385. All of the above results, except rurality, remained statistically significant following the Bonferroni correction.

## 4. Discussion

To our knowledge, this is the first study to investigate and report on the association between altitude of residence and the variation in cigarette smoking rates in the United States. County-level altitude data analyzed in this study suggest that regional altitude may significantly predict cigarette use while controlling for confounding variables such as race, rurality, educational attainment, physical inactivity, excessive drinking and mental distress, socioeconomic status, unemployment rate, and tobacco taxation [9,28,29,30,31,32,33,34,35,36,37,38]. We found that smoking prevalence is inversely associated with mean county altitude; specifically, for each 1000 foot increase in altitude, the smoking rate decreases by 0.143%.

When we examined ethnicity and race together with smoking prevalence, our findings aligned with prior reports that underscore higher smoking rates among Native Americans, Native Alaskans, and African Americans. Importantly, research has shown that individuals from these groups suffer an increased prevalence of smoking-related deaths from cardiovascular and lung diseases [40]. These healthcare disparities have been emphasized by the Centers for Disease Control and Prevention [41]. Thus, collectively, these findings highlight the need to further examine and address smoking disparities within specific ethnic groups and develop meaningful interventions to combat them. Data regarding educational attainment and ethnicity disparities in the U.S. show that lower education tends to be associated with increased cigarette use over time in all ethnic groups, and non-Hispanic whites have been shown to exhibit the highest cigarette consumption across all education levels over time [42]. Our findings also align with the extant literature in showing increased smoking with an educational attainment of high school or lower while controlling for other variables.

Multiple lines of research have reported increased smoking rates among people who live in poverty. For example, smoking prevalence is noted to be 41.1% among men with incomes below the federal poverty level, compared to 23.7% among those with higher incomes [43]. Moreover, more than a quarter of individuals below the poverty level are smokers, a rate that is twice that recorded in individuals that are above the poverty line. Similarly, our results also show that county smoking rates increase by 0.084% with each percentage increase in the poverty rate and by 0.122% with each percentage increase in unemployment. On the other hand, some studies have shown no relationship between tobacco consumption and being a student [44]. However, several studies have also demonstrated an inverse dose–response relationship between cigarette smoking and income level, poverty, and risky health behaviors [45], which overall underscores the need for smoking cessation efforts targeting the social determinants of health as a way to address disparities in the burden of disease and medical outcomes. 

As highlighted in Table 2, our results reveal a relationship between state taxation on tobacco and county smoking rates. Precisely, there was a significant decrease of −0.199% in county smoking rates with increased state taxation, a phenomenon which potentially may emerge as the most cost-effective strategy impacting youth or low-income groups. Notably, a 10% increase in tobacco product cost reduced its consumption in high-income countries by 4%. Therefore, leveraging tobacco taxes may be a powerful strategy to reduce smoking prevalence in different demographic groups; however, this implementation is limited on a global scale [46]. For example, in New Zealand, more robust tobacco control policies have led to significant declines in smoking prevalence compared to the United States, which may be attributed to higher cessation support programs and smoking-free laws [47].

Our finding showing that altitude displays an inverse association with cigarette smoking has face validity, which is provided by a canonical relationship in medicine: the cause-and-effect relationship between smoking and lung cancer, where an estimated 80–90% of lung cancer cases are linked to smoking [23,24]. Therefore, research suggesting that altitude is strongly and negatively associated with lung cancer incidence (*p* < 10^−16^) [20] lends plausibility to this study’s results. Additionally, atmospheric pressure, which decreases with altitude, is positively associated with lung cancer mortality [48]. Interestingly, both associations [20,48] are independent of smoking prevalence and survive testing for confounding and logical fallacy.

### 4.1. Pharmacokinetics of Inhaled Nicotine in Cigarette Smoke

The literature suggests the possibility that pharmacokinetic and pharmacodynamic factors may serve to reduce inhaled nicotine absorption from cigarettes, thus contributing to our finding that smoking decreases with altitude. The rapid delivery of nicotine to the brain through cigarette smoke inhalation is crucial for nicotine dependence. However, altitude introduces pharmacokinetic factors that may reduce inhaled nicotine absorption, potentially contributing to a reduction in smoking at higher altitudes. Additionally, due to the decreased atmospheric driving pressure of nicotine into the bloodstream at increased altitude, there is a well-described, albeit paradoxical, human physiological response to hypoxia: hypoxia induces vasoconstriction of the pulmonary vasculature. In the systemic vasculature, hypoxia causes a vasodilator effect through the adenosine-triphosphate-dependent potassium channel, leading to the relaxation of smooth muscle cells [49]. However, this does not apply to the lungs, where hypoxic pulmonary vasoconstriction is thought to be an intrinsic mechanism that optimizes systemic oxygen delivery by matching perfusion and ventilation [50]. By reducing perfusion of poorly ventilated areas of the lung, hypoxic pulmonary vasoconstriction decreases the shunting of desaturated, mixed venous blood to the systemic circulation [51]. In addition to vasoconstriction, there is evidence from human studies that altitude is associated with decreased lung volumes, i.e., forced vital capacity [52,53,54]. The effects of atmospheric pressure on both cigarette smoke inhalation and tobacco provide another potential explanation for why cigarette use might decrease with altitude. Increased draw resistance due to changes in atmospheric pressure at altitude may limit the amount of nicotine inhaled in cigarette smoke. However, given that tobacco combustion products cause cigarette addiction, cancer, lung disease, and heart disease [43,44]—and that combustion requires oxygen—it is at least theoretically possible that the reduced partial pressure of oxygen at altitude decreases combustion rates.

### 4.2. Pharmacodynamics of Nicotine and Altitude

Nicotine, analogous to other addictive substances, triggers central dopamine release from the nucleus accumbens, activating energy metabolism [55]. The primary binding sites for nicotine in the mammalian brain are α4β2 nicotinic acetylcholine receptors (nAChRs), particularly involving the β2 subunit. Nicotine-induced activation of α4β2 receptors increases dopamine release in the nucleus accumbens and prefrontal cortex, establishing the receptor’s critical role in nicotine addiction [56,57]. Positive allosteric modulation of α4β2 receptors, which has been demonstrated in preclinical studies, reverses nicotine withdrawal signs in mice [58]. Human trials of the α4β2 receptor partial agonist varenicline show positive results for smoking cessation [59]. Recent studies with novel α4β2 positive allosteric modulators indicate a reduction in nicotine self-administration, confirming the receptor as a valuable treatment target [60]. However, the hypobaric hypoxia that exists at even moderate altitude decreases oxygen availability, thus increasing oxidative stress and reactive oxygen species (ROS) [50]. Importantly, ROS are known to inactivate neuronal nicotinic acetylcholine receptors, irreversibly inducing long-term depression of neuronal currents [61]. Moreover, the α4β2 receptor, abundant in the mammalian brain, has been found to be specifically inactivated by ROS. Consequently, altitude-induced changes in α4β2 receptor signaling could potentially reduce nicotine use behaviors.

### 4.3. Mechanistic Pathways Involving Neurotransmitter Metabolism 

The pathways in nicotine addiction play a pivotal role in sustaining substance use disorders, particularly in smokers experiencing depressive symptoms modulated by the monoamine oxidase-A (MAO-A) enzyme. This is particularly relevant during withdrawal, for which dysphoria has been proposed as a key predictor of relapse. MAO-A, responsible for metabolizing serotonin and dopamine [62,63], exhibits increased activity in heavy smokers undergoing withdrawal [64], offering a mechanistic explanation for associated depressive symptoms. Moreover, smokers demonstrate a 40% inhibition of monoamine oxidase-B (MAO-B) in the brain [65]. As MAO-B is selective for dopamine, its inhibition increases dopamine functional availability, which contributes to reward signaling and addiction. Smoking dual inhibition of MAO-A and MAO-B enhances nicotine addictive potential. Evidence suggests that smoking reduces MAO-A and MAO-B binding sites in the brain, making the monoamine oxidases potential targets for smoking cessation [66]. However, altitude-induced hypoxia complicates this, as studies associate it with increased problematic substance use, potentially due to elevated addiction vulnerability resulting from MAO-B inhibition.

The notion that altitude may moderate smoking rates by reducing inhaled nicotine delivered by smoking cigarettes receives strong, albeit indirect, support from recent high-quality clinical trials of reduced-nicotine cigarettes. Despite historical concerns regarding smoking topography, i.e., the ability of smokers to maintain their usual nicotine dose and nicotine brain levels through unconscious compensatory changes in puff frequency and depth [67,68], the published human subjects data have motivated the U.S. Food and Drug Administration (FDA) to pursue a nicotine-limiting regulatory framework [69,70] that will reduce the nicotine content of cigarettes to “non-addictive levels” [71]. First, came a study published in the New England Journal of Medicine which reported that, in smokers with no intention to quit, reduced-nicotine cigarettes decreased nicotine exposure and nicotine dependence compared to standard cigarettes [72]. This was followed by a study of reduced-nicotine cigarettes that enrolled only participants from three demographic groups with increased vulnerability to tobacco addiction: patients with psychiatric disorders, opioid dependence, and socioeconomically disadvantaged women [73]. The investigators found that reduced-nicotine cigarettes demonstrate reduced addiction potential and reinforcing effects in all three vulnerable groups [73].

### 4.4. Developmental Considerations Relevant to Smoking, Altitude, and Adolescence 

Adolescence appears uniquely vulnerable to nicotine reinforcing effects, with significant initiation occurring before the age of 18. According to the Surgeon General’s report, the average age for daily smoking onset is approximately 18.3 years, emphasizing this period as critical for tobacco dependence [1]. Moreover, among adults who become daily smokers, the mean age of their first cigarette is 15.3 years, and the onset of daily smoking occurs at an average age of 18.3 years [1]. Furthermore, a cross-sectional study from Todorovic and colleagues in which they examined the prevalence of cigarette smoking amongst 1200 students and assessed factors related to tobacco use underscored that 74.9% of them had experimented with cigarette smoking, with up to 87% knowing about the harmful events of cigarette smoking [74]. These statistics suggest that adolescence may be a unique period of enhanced vulnerability to the reinforcing effects of nicotine [75]. 

Hypobaric hypoxia associated with altitude might contribute to lower smoking rates by effectively reducing inhaled nicotine doses, impacting the acquisition and maintenance of smoking during adolescence. Support for this hypothesis is found in both animal and human studies. For example, Schassburger et al. reported that adolescent rats self-administer low-dose nicotine less than adult rats, and that adolescents are less sensitive to the primary reinforcing effects of nicotine [76]. Meanwhile, clinical trial data indicate that low-nicotine cigarettes decrease nicotine withdrawal symptoms in adolescents, while producing reduced subjective positive effects, and, thus, may have a lower abuse liability [77]. In a multisite trial, young adults who were randomly assigned to the low-nicotine cigarette group reported lower satisfaction levels and smoked fewer cigarettes, suggesting reduced smoking reinforcement [78]. In summary, the data align with the notion that altitude may decrease systemic nicotine delivered by inhaled tobacco smoke, offering a potential avenue for reducing smoking prevalence in adolescents.

### 4.5. Limitations 

Several limitations should be taken into consideration when interpreting this study results. For instance, individual differences related to smoking behaviors, including the frequency of puffs, the depth and duration of inhalation, or the length of time smoke is held in the lungs, could influence nicotine consumption at higher altitudes. Future research should assess differences in smoking topography in a controlled environment to determine if altitude affects nicotine pharmacokinetics and pharmacodynamics. Another potential limitation includes the possibility of underreporting cigarette use to avoid judgment, especially in regions where smoking is not culturally acceptable. Also, at this point, it remains unclear how the introduction of ‘vaping’ has impacted nicotine usage in the U.S. and what percent of smokers use both cigarettes and e-cigarettes. Although we did not include e-cigarettes and vaping data in the current study, some studies have estimated that more than 68 million people vaped worldwide in 2020 [79]. This is a concerning aspect considering that vaping could lead to nicotine use disorders and further perpetuate these behaviors; for example, a study of more than 3900 students has shown that vaping frequency was associated with higher odds of smoking, underscoring the need to focus on these policies in the upcoming years. Future analyses should examine the confounding variables and complexity introduced by the emergence of the ‘vape’ industry.

## 5. Conclusion

Overall, this study expands the existing literature by reporting an inverse association between the prevalence of cigarette smoking and altitude. We have described, briefly, several interrelated mechanisms by which inhaled nicotine in cigarette smoke may result in decreased addiction as altitude increases and the ambient atmospheric pressure decreases, such as pharmacokinetics, pharmacodynamics, and mechanistic pathways.

In addition, we have proposed potential factors and developmental contributors to this association, which may guide targeted interventions and increase awareness to reduce the public health impact of smoking, relating to decreasing atmospheric pressure as altitude increases and the human physiological response to hypobaric hypoxia.

Future investigations are needed to replicate these results and to further explore the relationship between altitude and smoking behavior beyond the United States. Moreover, future studies should examine how altitude may represent a protective factor for cigarette smoking. Identifying patterns in cigarette use may provide valuable information that could enable national organizations and legislative bodies to establish effective tobacco control initiatives. Implementing successful tobacco prevention and cessation programs may assist in lowering the prevalence of cigarette use on a national level and lead to a decrease in smoking-related diseases and premature death on a global scale.

In conclusion, to our knowledge, this study is the first to report a negative association between the prevalence of cigarette smoking and altitude. In addition to the face value of this result conferred by the replicated finding that lung cancer rates are negatively correlated with altitude, we have identified potential pharmacokinetic, pharmacodynamic, mechanistic pathway, and developmental contributors to this association. In addition, this finding converges with evidence from multiple clinical trials of reduced nicotine cigarettes, a fact that has motivated the FDA to consider regulatory action and may ultimately reduce the public health impact of smoking.

## Figures and Tables

**Figure 1 ijerph-21-00226-f001:**
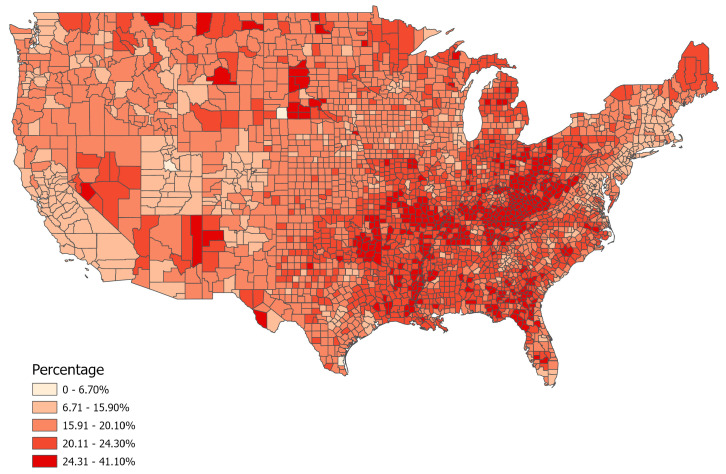
County-level prevalence of cigarette smoking in the U.S. 2020, (*N* = 3106).

**Figure 2 ijerph-21-00226-f002:**
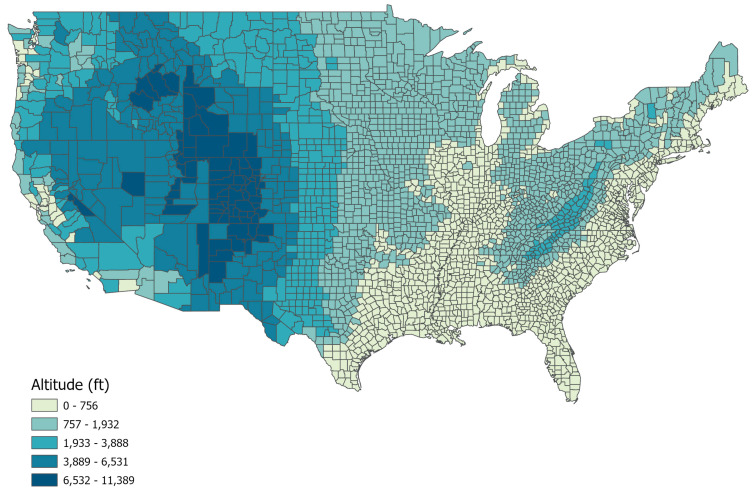
Mean county-level in the United States 2020, (*N* = 3106).

**Figure 3 ijerph-21-00226-f003:**
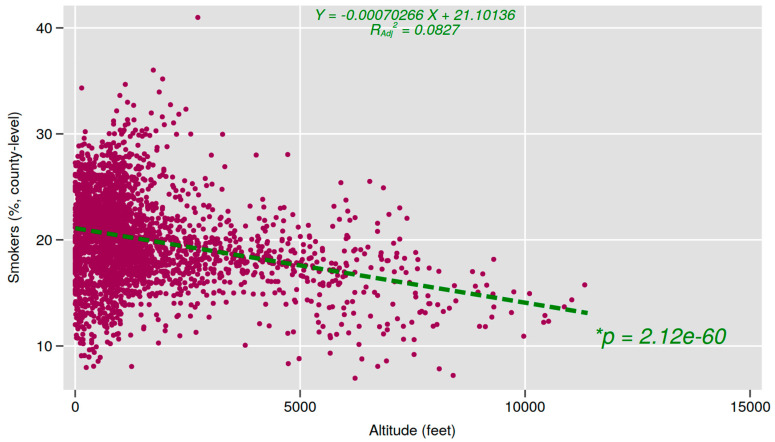
County-level smoking by altitude in the United States, 2020. * Significant county-level smoker percentage differences by altitude in the United States.

**Table 1 ijerph-21-00226-t001:** Summary of selected 2020–2021 county-level variables.

	No.	Mean	Median	SD	Min	Max	Pearson Correlation with % Smoking	*p* Value
**Environmental**								
Altitude (ft)	3106	1449.21	912.01	1665.62	2.63	11388.98	−0.291	**
**Demographic**								
% Female	3106	49.59	49.96	2.28	24.56	57.05	−0.054	*
% American Indian/Alaskan native	3106	2.17	0.71	6.61	0.00	85.68	0.207	**
% African American	3106	9.13	2.41	14.27	0.00	85.62	0.165	**
% Non-Hispanic white	3106	75.51	82.60	19.92	2.68	97.59	0.069	**
% High school degree	3106	87.94	89.21	5.87	49.67	99.40	−0.525	**
% Rural	3106	63.93	66.52	33.56	0.00	100.00	0.435	**
**Health**								
% Physically inactive	3106	25.74	25.20	5.18	11.30	47.20	0.779	**
% Excessive drinking	3106	19.07	18.85	3.22	8.19	28.93	−0.371	**
% Frequent mental distress	3106	15.76	15.70	2.02	10.10	23.30	0.779	**
**Socioeconomic**								
% Poverty	3106	13.75	12.80	5.40	3.00	43.90	0.661	**
% Unemployed	3106	4.62	4.40	1.70	0.89	17.30	0.180	**
State excise tax	3106	1.34	1.20	0.95	0.17	4.50	-0.323	**

** Significant at *p* < 0.001; * Significant at *p* < 0.05.

**Table 2 ijerph-21-00226-t002:** Multivariate linear regression analyses of predictors for smoking.

	β	Lower	Upper	*p* Value
Mean county altitude (× 10 ^3^, or per 1000 feet)	−0.143	−0.178	−0.108	**
% Female	−0.109	−0.132	−0.087	**
% American Indian/Alaskan native	0.176	0.167	0.186	**
% African American	0.049	0.043	0.055	**
% Non-Hispanic white	0.113	0.108	0.118	**
% High school degree	−0.070	−0.085	−0.055	**
% Rural	0.003	0.001	0.005	*
% Physically inactive	0.410	0.389	0.431	**
% Excessive drinking	0.078	0.057	0.098	**
% Frequent mental distress	0.518	0.479	0.558	**
% Poverty	0.084	0.067	0.100	**
% Unemployed	0.122	0.086	0.158	**
State excise tax	−0.199	−0.260	−0.138	**

** Significant at *p* < 0.001; * Significant at *p* < 0.05.

## Data Availability

The data presented in this study are available on request from the corresponding author.

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
