# Peer review of "Exploring the Link between Altitude of Residence and Smoking Patterns in the United States"

_ijerph, 2024, doi:10.3390/ijerph21020226_

Round 1

Reviewer 1 Report

Comments and Suggestions for Authors

Review IJERPH-2808415

Title: Exploring the Link Between Altitude of Residence and Smoking Patterns in the United States

Comments from reviewer

Overall, this manuscript is to examine the potential relationship between altitude and the prevalence of cigarette use at the county level (N=3,108), across the continental U.S. They used the multivariate linear regression and found that smoking rates were negatively associated with altitude. The authors suggest that altitude may affect pharmacokinetics and pharmacodynamics involved in cigarette use. I have some major comments on your manuscript. Overall, it is an interesting paper, but I have some major and minor comments regarding the contents of your paper. Especially, some of sections (e.g. introduction, method, and result sections) need to be restructured and recreated and significantly expanded to support your scientific claims. Detail explanations are listed in below:  

1)     In the introduction section, please find any previous literatures to introduce potential association between altitude and smoking if any.

2)     In the method section, you used 2014 Smoking rate data based on 2016 County Health Ranking National Database. Have you thought about using more recent data as CHRND publishes county ranking national information annually? I think 2022 or 2023 database is now available and I personally think 2014 data is a bit old to use. Please update. 

3)     In the method section, it didn’t fully explain how the mean county altitude data was calculated. For example, what method did you apply to obtain mean county altitude for each county from Shuttle Radar Topography Mission (SRTM)? What year of SRTM data was used? Please elaborate.

4)     In the method section, same comment as #1, if you have more recent data on smoking rate data, then more recent potential covariates need to be used for the sake data consistency.

5)     In the method section, you included 3 different races in your statistical model, and I strongly believe it’d result in multicollinearity in the model. For the sake of curiosity, have you tested potential multi-collinearity problem between these covariates such as testing Variance Inflation Factor (VIF) or correlation matrix? If so, did you find any high correlations between any variables?  How do you plan to adjust the issue if exists? Please elaborate.

6)     In the method section, on potential covariates, there should be more socio/demographic/economic/health-clinic factors, which are closely related to smoking rate, and I believe you’ll need to include more of them to build more reliable statistical model. You only included 4 factors excluding racial factors.

7)     Based on rebuilding and updating the model, discussion should be updated accordingly. That is, your current model structure has a significant drawback, which should be adjusted and corrected before making a conclusion on your claim.

8)     In the method section, please create maps on smoking rate and altitude on 3,108 counties, so that readers can better understand visually on the regional variations of the variables. Please use GIS software to create them.

Comments on the Quality of English Language

Moderate editing of English language required. 

Reviewer 2 Report

Comments and Suggestions for Authors

The paper is very well written, and I only have some minor suggestions to improve the quality of the paper.

1. Page 2 of 15

Line : 52 

"from 20.9% of the population in 2005 to 16.8% in 2014"

Authors may consider using the most recent data on tobacco use:

"In 2021, 11.5% of U.S. adults currently smoked cigarettes"

Source:https://www.cdc.gov/tobacco/data_statistics/fact_sheets/fast_facts/index.htm

2. Page 3 of 15

Line 100: BRFSS 

Line : 128: American Community Survey and the EDfacts datasets.

It is recommended that authors provide a brief explanation or demonstration of the relationship between CHRND, BRFSS, American Community Survey, and EFfacts datasets. It appears that the authors utilized or possibly merged multiple datasets, and I am unclear about the processing methodology employed with these diverse datasets.

3.  Page 5 of 15

Discussion

Line : 180

"County-level altitude data analyzed in this study suggests that regional altitude is a significant predictor of cigarette use"

Is the study design an ecological study due to the unit is county-level, or is it cross-sectional? The authors may want to consider adding one or two sentences to clearly indicate the study design.

4. Page 6 of 15

Line : 218

"this implementation is limited on a global scale. "

Authors may want to add some examples that demonstrate or highlight the limited implementation on a global scale, such as making comparisons between the United States and other countries.

5. The Discussion section is very comprehensive and well-written.

Round 2

Reviewer 1 Report

Comments and Suggestions for Authors

Thanks for addressing my comments and the paper has been improved throughout. I have no further comments except there might be one more round of minor English changes and spell checks recommended.

Comments on the Quality of English Language

Thanks for addressing my comments and the paper has been improved throughout. I have no further comments except there might be one more round of minor English changes and spell checks recommended.